# Pulmonary Vein Stenosis Associated with Germline *PIK3CA* Mutation

**DOI:** 10.3390/children9050671

**Published:** 2022-05-05

**Authors:** Delphine Yung, Kaitlyn Freeman, Ghayda Mirzaa

**Affiliations:** 1Department of Pediatrics, Division of Cardiology, University of Washington School of Medicine, Seattle, WA 98195, USA; kaitlyn.freeman@seattlechildrens.org; 2Center for Integrative Brain Research, Seattle Children’s Research Institute, Seattle, WA 98105, USA; ghayda.mirzaa@seattlechildrens.org; 3Department of Pediatrics, University of Washington School of Medicine, Seattle, WA 98195, USA; 4Brotman Baty Institute for Precision Medicine, Seattle, WA 98195, USA

**Keywords:** pulmonary vein stenosis, *PIK3CA*, alpelisib

## Abstract

Pulmonary vein stenosis is a rare and frequently lethal childhood disease. There are few known genetic associations, and the pathophysiology is not well known. Current treatments include surgery, interventional cardiac catheterization, and more recently, medications targeting cell proliferation, which are not uniformly effective. We present a patient with PVS and a *PIK3CA* mutation, who demonstrated a good response to the targeted inhibitor, alpelisib.

## 1. Introduction

Pulmonary vein stenosis (PVS) is a critical narrowing of one or more pulmonary vein(s) that affects an estimated 1.7 per 100,000 children under the age of two years [1]. Risk factors for the development of PVS include congenital heart disease, especially total anomalous pulmonary venous connection, prematurity with bronchopulmonary dysplasia (BPD), and select genetic syndromes. In some children, the stenosis can be improved with surgery, such as sutureless marsupialization repair, or catheter-based interventions, such as balloon dilation and stenting. However, the disease often becomes recurrent and progressive, sometimes leading to complete vein atresia, and results in pulmonary hypertension, heart failure, and death [2,3,4,5]. Mortality rates are up to 83% in patients with three or more stenosed veins [6].

The etiology of PVS is poorly understood, but histopathological examination of the vessel lesions demonstrates neo-intimal hyperplasia and proliferation of myofibroblasts, and immunohistochemistry demonstrates strong receptor tyrosine kinase expression [7,8]. To delay disease progression, drug therapies directed at pathways related to cellular proliferation have been trialed over the past decade [9]. Current targeted medical treatments in clinical use include losartan (transforming growth factor beta pathway) [10], imatinib (tyrosine kinase inhibition), bevacizumab (vascular endothelia growth factor A inhibition) [11], and sirolimus (mammalian target of rapamycin (mTOR) pathway) [12,13,14]. We present a case of a term infant with congenital heart disease and recurrent PVS who was found to have a mutation in the phosphatidylinositol 3-kinase protein kinase B mTOR (PI3K–AKT–mTOR) pathway, and who was successfully treated with alpelisib, an inhibitor of this pathway.

## 2. Case Presentation

A three-week-old female presented with respiratory distress, hypotonia, rapid head growth and cardiomegaly on chest X-ray. She was born at term via c-section with a birthweight of 3.7 kg (80 percentile) due to failure to progress after an unremarkable pregnancy. She spent five days in the neonatal intensive care unit for hypoglycemia and respiratory distress requiring a high flow nasal cannula. The parents brought her to the pediatrician two weeks after discharge for poor feeding and lethargy.

On admission, the echocardiogram showed a large perimembranous ventricular septal defect (VSD) and normal pulmonary veins without stenosis. The head ultrasound revealed multiple small bilateral ventricular cysts and further brain MRI showed megalencephaly with diffuse hypo/dysmyelination consistent with megalencephalic leukodystrophy. A renal ultrasound showed scattered echogenic foci in the upper and interpolar medullary pyramids. Bloodwork was normal except for indirect hyperbilirubinemia. A genetics evaluation was performed, and diagnostic exome sequencing was sent. The patient improved with furosemide therapy and was discharged home after seven days with feeding therapy.

At the three-week follow up, the patient was well with good weight gain and no respiratory distress. However, the echocardiogram showed new evidence of left lower pulmonary vein (LLPV) stenosis with a mean gradient of 6 mmHg. A CT angiogram confirmed narrowing of the common trunk of the left-sided pulmonary veins (Table 1). The following week, oral feeds had decreased, work of breathing had increased, and oxygen saturation was 82%. A repeat echocardiogram revealed the LLPV gradient had increased to 12 mmHg and new pulmonary hypertension with bidirectional shunting. She underwent surgical VSD closure and left-sided pulmonary vein sutureless repair (Figure 1). She also underwent a complete evaluation for pulmonary hypertension, including a feeding evaluation that diagnosed aspiration, and a sleep study that diagnosed obstructive sleep apnea. Oral feedings were stopped, and she was discharged with tube feeding and oxygen. A postoperative echo showed no residual gradient through the left pulmonary veins and normal pulmonary pressure.
children-09-00671-t001_Table 1Table 1Patient parameters at time of PVS diagnosis.General
Body weight4.8 kgAge2 monthsSexFemaleEthnicityAsian (Korean)**Echocardiogram**
Pulmonary vein gradient5.6 mmHgRVDilated with normal systolic functionLVNormal size and functionTR gradient3.9 m/s TR jet in setting of large VSDInterventricular septal positionFlattenedAtriaNormal size**Electrocardiogram**
RhythmNormal sinus rhythmHeart rate146 per minuteQRS duration70 msPR interval100 msQTc interval451 msT waveNormal
Figure 1Timeline of treatments.
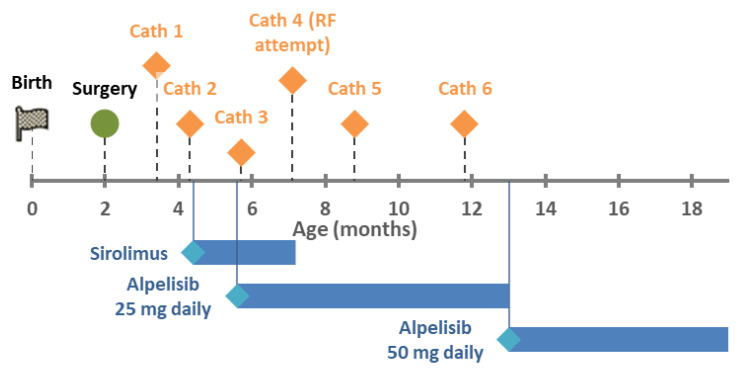


Three weeks post-discharge, an echocardiogram showed LLPV re-stenosis with a mean gradient of 12 mmHg. The RV pressures remained normal with a TR jet velocity of 2.7 m/s. Over the next six weeks, she underwent two catheter-based left pulmonary balloon venoplasties for progression of venous stenosis with gradients over 9 mmHg. She was started on sirolimus with a target level of 6–10 mcg/L. Despite this, the LLPV progressed to atresia by her third interventional catheterization, and a fourth catheterization was planned for the purpose of re-canalization. In the interim, she was admitted for urosepsis. After that, the fourth catheterization proceeded, but was unsuccessful and resulted in perforation of her descending aorta. This was further complicated by culture negative sepsis, and sirolimus was stopped.

Trio exome sequencing revealed a pathogenic germline *PIK3CA* mutation (c.1345C > T, p.Pro449Ser). She was initiated on alpelisib at five months of age. The fifth and sixth catheterizations occurred over the following five months at progressively increasing intervals for echocardiographic gradients over 9 mmHg. There have been no further catheterizations at nineteen months of age. Echocardiograms demonstrated a stable LUPV gradient of 7–8 mmHg and TR jet velocities between 2.7–3.0 m/s.

Developmentally, she has made remarkable progress despite her initial neurological evaluations. At one year, she was able to sit unassisted, and at 18 months, she could stand and cruise as well as smile, clap, babble, and sign. A swallow study at ten months showed significant improvement and thin liquids were re-introduced. A repeat brain MRI at 14 months of age showed interval myelination compared to her study at 1 month and no cortical malformations. She was weaned off nighttime oxygen after tonsillectomy and adenoidectomy at 15 months of age.

## 3. Discussion

### 3.1. Case Discussion

Like many patients with severe PVS, this patient presented as a neonate with congenital heart disease but initially normal pulmonary veins and underwent surgical repair, which was not curative. She also had a large left-to-right shunt, which in certain patients has been demonstrated to increase the risk for recurrent PVS, and may prove to be an independent risk factor [15]. In addition, she was not diagnosed or treated for aspiration, another known risk factor for PVS [16], until after surgery. Pulmonary hypertension was first noted due to bidirectional shunting across the VSD at the same time as PVS was first seen. Fortunately, after VSD closure and treatment for aspiration and sleep apnea, pulmonary hypertension has been mild. Pulmonary hypertension has remained mild in the setting of residual PVS, and no pulmonary hypertension medical therapy has been indicated. Other markers of initial severe disease included symptoms of respiratory and cardiac failure, and rapid progression of one pulmonary vein to atresia. Additionally, the brain findings carried a poor neurodevelopmental prognosis.

Given the child’s complex clinical presentation with multi-system involvement, trio exome sequencing (ES) was performed. ES is a good first- or second-tier test for individuals with congenital anomalies, as recommended by the College of Medical Genetics and Genomics [17]. Knowledge of genetic conditions can contribute to treatment; for example, patients with Trisomy 21 and Smith–Lemli–Optiz syndrome are known to have rapid progression of PVS, and in Trisomy 21, a higher risk for pulmonary hypertension and mortality, so early treatment should be aggressive [5]. In this case, the genetics revealed both a potential pathophysiology and a novel treatment option, showing the benefit of increased genetic testing in patients with PVS. As is also common, she underwent surgical intervention initially, followed by catheterization-based therapy. The usual standard of care is to limit catheter interventions to balloon angioplasty due to the complications of stent placement in infants [18]. The unfortunate rapid progression of the LLPV to atresia led to the decision for stent placement in the LUPV when stenosis started to progress.

Over the past several years, medical therapy of PVS has become more common since reports of imatinib and sirolimus demonstrated improved mortality. After our patient had rapid progression leading to the second catheterization, we elected to add medical therapy to delay progression. Sirolimus was initially chosen over imatinib due to overlap in known *PIK3CA* mutation and the mTOR pathway (see Section 3.2 below). However, there was no obvious improvement in PVS since two further catheterization procedures were performed over the following two months and the LLPV progressed to atresia. In addition, attempts at blood draw to check sirolimus levels became difficult and were sometimes unsuccessful. Sirolimus was continued after initiation of alpelisib due to the unknown efficacy of alpelisib but stopped after three months due to frequent serious infections, which did not recur after discontinuation.

Alpelisib was started after a delay in obtaining the medication (see Section 3.3 below). Discounting cath #3, which was performed two days after initiation, and cath #4, which was not performed due to interval worsening, there have only been two cardiac catheterization procedures—at 3 and 6 months after initiation—with 13 months of stability after that. In addition, she appears to have made neurodevelopmental improvements as well. There have been no side effects to the drug, to date. Thus, alpelisib appears to be associated with a significant improvement in her PVS phenotype.

### 3.2. PIK3CA

#### 3.2.1. PI3K–AKT–mTOR Signaling Pathway

The PI3K–AKT–mTOR pathway regulates cell proliferation and overgrowth (Figure 2).

#### 3.2.2. *PIK3CA* Mutations and Clinical Phenotypes

*PIK3CA* mutations cause constitutive activation of the PI3K–AKT–mTOR pathway leading to cell proliferation and overgrowth [19]. Mutations most commonly arise post-zygotically, are exclusively activating (never loss of function), and concentrated at one of several mutational “hotspots”. Mutations are well known to cause cancer in adults [20]. In children, *PIK3CA* mutations in similar locations cause *PIK3CA* related overgrowth spectrum (PROS). PROS describes a growing number of phenotypes featuring congenital overgrowth with hyperplasia in a wide variety of tissues including brain, body, and vessels [20,21].

Congenital heart defects have been reported in children with *PIK3CA* related megalencephaly with capillary malformations (MCAP), including ventricular septal defect, atrial septal defect, patent ductus arteriosus, persistent left SVC, aortic arch anomalies, plus tachyarrhythmias [22]. Germline *PIK3CA* mutations are rare, and previously were considered lethal in the embryonic stage. In our patient, the constellation of the child’s features, including megalencephaly, white matter abnormalities, and hypotonia, broadly fit within PROS. Further, the *PIK3CA* variant identified in this child has been previously reported in individuals with features of PROS as well, supporting a causal association. Please see Figure 3 for gene structure and mutation location.

### 3.3. Alpelisib

Alpelisib (BYL719) is a select PI3K inhibitor that has been recently approved for *PIK3CA*-mutated hormone receptor positive breast cancer and has been used in children with somatic features of PROS [23]. Known side effects include glucose dysregulation and rash. It is commercially available due to the approval for adults with breast cancer. In children, it is being compassionately studied in those older than 2 years with PROS (ClinicalTrials.gov Identifier: NCT04980833 and NCT04085653). Because our patient was under age 2 years, she was not eligible for research studies, and despite discussion with the manufacturer, she was not eligible for compassionate use. She finally was able to obtain the medication with private insurance authorization and appeal. The enteral dosage for ages 2–18 is 50 mg daily. We started with 25 mg daily and increased to 50 mg daily at 13 months of life.

### 3.4. Future Directions

We hypothesize that *PIK3CA* mutations could play a role in pulmonary vein stenosis. The pathophysiology of PVS could be considered a form of recurrent local “overgrowth” or localized hyperplasia, like other known phenotypes of *PIK3CA* mutations. Identification of mosaic mutations as a cause of PVS would be paradigm-shifting in the care of affected children. If *PIK3CA* mutations are identified, alpelisib may be a consideration as a targeted therapy to prevent the progression of PVS. This highlights the importance of multi-disciplinary care including genetics, and multi-center studies to advance PVS knowledge.

## 4. Conclusions

This is the first report of a patient with PVS and a *PIK3CA* mutation who appears to have responded to a targeted inhibitor, alpelisib. Future research should investigate whether *PIK3CA* mutation contribute to other patients with PVS and whether alpelisib may be a useful medical therapy.

Main concern: PVS is a rare and frequently lethal childhood disease with few known genetic associations or treatments.Main discovery: We present a patient with PVS and a *PIK3CA* mutation, who demonstrated a good response to the targeted inhibitor, alpelisib.Main take-home message: Future patients with PVS may benefit from further research with whole exome sequencing and alpelisib therapy.

## Figures and Tables

**Figure 2 children-09-00671-f002:**
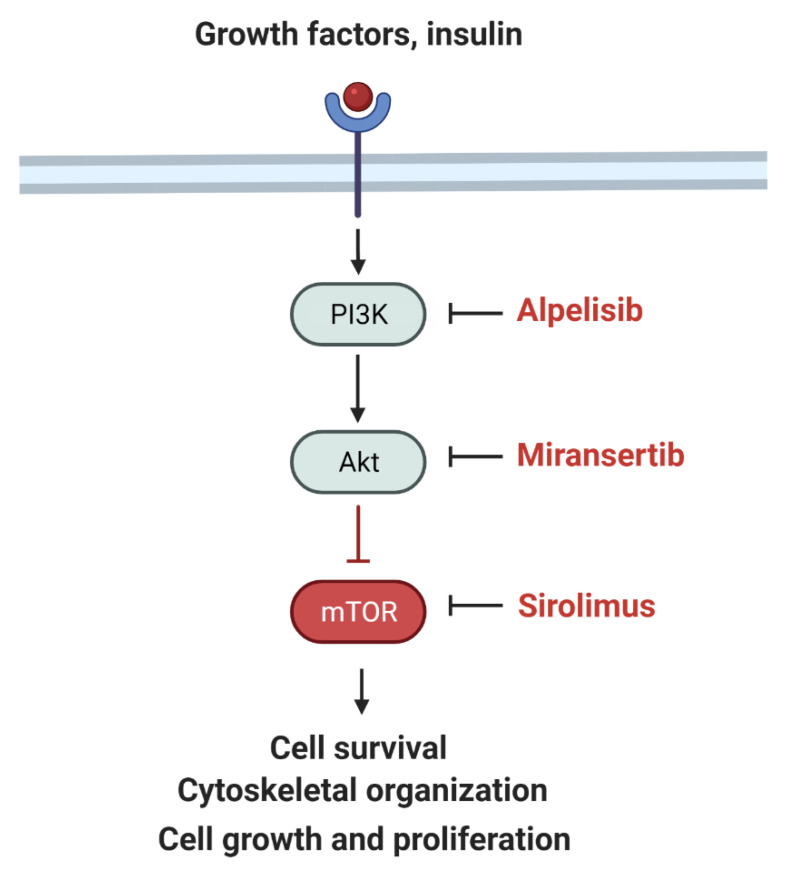
Simplified schematic of the PI3K signaling pathway and inhibitors including alpelisib (PI3K inhibitor), miransertib (AKT inhibitor), and sirolimus (MTOR inhibitor). Adapted from “mTOR signaling pathway” by BioRender.com (2022). Retrieved from https://app.biorender.com/biorender—accessed on 15 March 2022.

**Figure 3 children-09-00671-f003:**
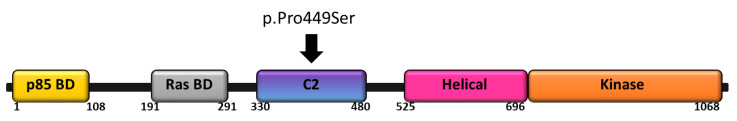
Schematic representation of *PIK3CA* including known functional domains. The c.1345C > T (p.Pro449Ser) variant that lies within the C2 domain is shown.

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
