# Peer review of "Pulmonary Vein Stenosis Associated with Germline PIK3CA Mutation"

_children, 2022, doi:10.3390/children9050671_

Round 1

Reviewer 1 Report

The article titled: << Pulmonary Vein Stenosis Associated with Germline PIK3CA Mutation >> is authored by Yung D et al.

The authors are presenting a Case Report about a patient with PVS, who have shown positive response to treatment with Alpelisib, an inhibitor of PI3K-AKT-mTOR pathway. The patient had a PIK3CA mutation.

The paper is of great interest. However, some concerns could be addressed prior to publication.

  1. Provide a table displaying some parameters about the patient:

General: Body weight, age, sex, and ethnicity.

Cardiac echography: Pulmonary vein hemodynamic, myocardium metrics (LV, RV, septum, atria).

ECG: p wave, PR, QRS, T wave, heart rate at rest.

  1. Add a displayed item (table or figure) summarizing the Alpelisib posology that has been used (with success) for this patient.

  1. Provide a schematic of the genes’ structure and the suspected (or confirmed: supported by references) location of the PIK3CA mutation.

  1. Add a Highlight section with up to 4-bullet points, stating the main concern, the main discovery, and the main take-home message of the paper.

Reviewer 2 Report

To the editor:

The manuscript by Yung et al. “Pulmonary Vein Stenosis Associated with Germline PIK3CA 2 Mutation” present a patient with PVS and a PIK3CA mutation, 13 who demonstrated a good response to the targeted inhibitor alpelisib. We are in an information growth around the disease of pulmonary vein stenosis. It is important for cases like this case report and discussion to be presented to the larger community of providers caring for these unique children. The case report is well written, and I commend the authors. The authors present an observation and provide a plausible biological explanation for the link between the genetics and the disease

The strengths of this manuscript are significant but offset by some concerns described below.  General and specific comments follow by section.

GENERAL COMMENTS

  1. How do we know that the VSD alone did not lead to the PVS?
  2. Did the authors discuss or reference the association of aspiration with PVS?
  3. Was there significant PH? and What was the progression of the PH?

COMMENTS BY SECTION

I.ABSTRACT: Clear, concise, and well written

  1. INTRODUCTION. The introduction provides A) background; B) the question that needs to be answered/the clinical significance of work; and C) the main objectives of the study. The authors touch upon each of these features. One comment
  2. I would suggest adding the word “term” before infant on line 35.

III. CASE PRESENTATION: Well presented. A few additional details could enhance the presentation

  1. Please provide the birthweight the centile (IUGR/SGA and VSD can lead to NEC and can also be associated with PVS)
  2. The Figure is nicely presented and allows the reader to follow the progression of the case’s timeline

IV. DISCUSSION: The discussion has all the major elements to provide the most clear and comprehensive addition to the literature. A few additional comments

  1. The authors could speak about the progression of disease and PH as seen in other genetic conditions (e.g. T21, SLO?) Infants with Smith-Lemli-Opitz are often diagnosed earlier, between birth and 2 months of life, and have a clinical course that rapidly progresses from single vein to multi-vein involvement. PVS in infants with Trisomy 21 has been found to be rapidly progressive, often associated with PH, and carries a high mortality rate within a few years of diagnosis. The progression of PVS in Trisomy 21 patients appears to be more aggressive than other neonates. Similarly, Trisomy 21 infants are also at an increased risk of having more rapid progression of PH compared to the general population, in part due to abnormalities in pulmonary vasculature development, impairments in regulation of vascular tone and vascular endothelial function.

V. FIGURES AND TABLES
Figure 1: Clear

Figure 2: Clear

VI. REFERENCES. The reference section is appropriate. One other manuscript to consider

  1. PMID: 34572215

Round 2

Reviewer 1 Report

The authors perfectly answered most reviewers' comments, however, the Figure 3, mentioned by the authors in their response document is not visible in the manuscript.

Please add figure 3.

Author Response

We apologize for the difficulty in viewing Figure 3. We will upload it directly this time.